# Push-Poke: Collision based Direct Manipulation Technique for Plane Alignment in Virtual Reality

Sriram Kumar*
Faculty of Information Technology
and Communication Sciences
Tampere University

Jari Kangas†
Faculty of Information Technology
and Communication Sciences
Tampere University

Helena Mehtonen‡
Medical Imaging Centre,
Department of Radiology,
Tampere University Hospital

Jorma Järnstedt§
Medical Imaging Centre,
Department of Radiology,
Tampere University Hospital

Roope Raisamo¶
Faculty of Information Technology
and Communication Sciences
Tampere University

## ABSTRACT

Medical operation planning requires high precision due to the health risks involved. In virtual reality-based osteotomy operation planning, medical professionals would like to use their hands instead of the controllers for osteotomy plane manipulation. However, using hands as an input method in virtual reality is challenging due to noisy hand tracking. We explored the perceptual structure of precise plane manipulation by conducting a controlled experiment to compare: (1) separable Push-Poke which dynamically selects plane manipulation parameters, (2) separable custom axis with Control-Display gain widget which provides user to select object manipulation parameters, (3) integral pinch-based direct manipulation. The perceptual structure of hand-based plane manipulation techniques is composed of (1) integral and fast direct manipulation and (2) separable slow technique that dynamically selects manipulation parameters.

**Index Terms:** Human-centered computing—Human computer interaction (HCI)—Interaction paradigms—Virtual reality; Human-centered computing—Human computer interaction (HCI)—Interaction techniques—Gestural input; General and reference—Cross-computing tools and techniques—Empirical studies

## 1 INTRODUCTION

Osteotomy is a surgical incision procedure performed by medical professionals to shorten, lengthen, or change the position and orientation of bones [13]. Jaw osteotomy surgery is performed for roughly 5% of the world population for problems such as jaw misalignment (a receding chin, open bite), TMJ (temporomandibular joint) disorder, sleep apnea, and malocclusion problems [46]. Operation planning is used to reduce the high health risks of around 10%-20% [47]. In jaw osteotomy planning, the jaw cutting process is planned in three steps: (1) marking points to create an initial osteotomy plane, (2) manipulating the position, orientation, and scale of the osteotomy plane, and (3) cutting the jaw with the osteotomy plane. These three steps are repeated to do more complex procedures.

Currently, medical professionals plan jaw osteotomy using a traditional two-dimensional (2D) screen-based user interface with a keyboard and mouse. It requires high cognitive load to mentally reconstruct the 3D anatomical structure from 2D and it is also time-consuming, error-prone and requires training. In comparison,

---

*e-mail: sriramkumar.kishorekumar@tuni.fi

†e-mail:jari.a.kangas@tuni.fi

‡e-mail:helena.mehtonen@pshp.fi

§e-mail:jorma.jarnstedt@pshp.fi

¶e-mail:roope.raisamo@tuni.fi

viewing 3D medical data in a 3D virtual reality (VR) environment reduces the 3D to 2D information loss, provides enhanced depth perception due to stereo display and motion parallax (due to small and frequent head motions) and ability to interact with two hands simultaneously [41]. In 2D user interfaces, points are marked on the skull to reduce the plane manipulation effort in 2D, while for VR, the enhanced depth perception allows users to proceed with plane manipulation step in 3D without the need for marking points.

A user expectations study was conducted with two medical professionals with jaw osteotomy experience and they performed the plane manipulation step in VR using controllers. The medical professionals were able to complete the task precisely and quickly. However, they expected to be able to manipulate the plane with their bare hands instead of controllers as hand manipulation was more inline with their mental model which has also been observed in other studies with medical users [10, 26]. These usability and user experience studies [10, 25, 26, 53] with medical users have used pinch as the hand-based interaction technique. User studies [7, 18, 22] have found that hand based pinch interaction has lower object manipulation precision than controllers which is mainly due to noisy camera sensor data caused by egomotion of the head, lack of FOV, occlusion, illumination, and background noise [40].

Previous hand-based interaction techniques have shown to reduce the effect of this noisy camera sensor data on the precision of object manipulation task by designing hand-based object manipulation techniques using different design factors such as direct manipulation, indirect manipulation, Rotation Translation (RT), Degrees of Freedom (DoF), Control-Display (CD) gain, transformation axes, rotation pivot point and feedback. However, some of these techniques [3,35,36,57] took more time to complete the task as these did not match the users' mental model. Jacob et al. [23] have shown that matching the perceptual structure of the interaction technique with the perceptual structure of the task increases the precision of the task and reduces the task completion time. Therefore, this research focuses on exploring design factors of hand-based object manipulation techniques that would match the perceptual task structure of precise osteotomy plane alignment task in VR and evaluates whether these interaction techniques can reduce the effect of the noisy camera sensor in comparison to pinch.

In the design process, a contextual inquiry was conducted to understand how medical professionals adjusted the osteotomy plane in the osteotomy operation planning process. Based on this contextual inquiry, a plane alignment task was created for the study. From reviewing existing object manipulation interaction techniques, we found that separable interaction techniques [31,35,36] that provided users the ability to select rotation and translation (RT) and Degrees of Freedom (DoF) were able to achieve higher precision. Based on the taxonomy, we created separable interaction techniques that would allow users to select these parameters or dynamically selected

based on user interaction: (1) Push-Poke and (2) custom axis with Control-Display gain (CACD). A controlled experiment with 12 participants was conducted with pinch-based direct manipulation as the integral interaction technique to identify which interaction technique matched the perceptual structure of the plane manipulation task.

The results of the controlled experiment showed Push-Poke was objectively and subjectively more precise than baseline pinch. Push-Poke was preferred to pinch and CACD because it was intuitive, easy to use and participants felt confident in using it. Based on the result, the perceptual structure of hand-based plane manipulation techniques is composed of (1) integral and fast direct manipulation and (2) separable slow technique that dynamically selects manipulation parameters. In future work, Push-Poke technique could be evaluated with medical professionals for the jaw osteotomy operation planning process and compared with controllers to understand whether Push-Poke could potentially replace controllers.

In summary, our contributions are: (1) proposed two separable plane manipulation techniques to support user selection or dynamic selection of plane manipulation parameters, (2) empirical validation of plane manipulation techniques, and (3) understanding of perceptual structure of hand-based precise plane manipulation task.

The structure of this paper is as follows: Section 2 discusses the related work, Section 3 explains the design process followed to design interaction techniques for jaw osteotomy plane manipulation with different perceptual structures, Section 4 states the research questions addressed in this paper, Section 5 describes the study to find the perceptual structure of the interaction technique to match the plane manipulation task, Section 6 presents the results of the study, Section 7 explains the perceptual structure of hand-based plane manipulation in VR based on the results of the study, Section 8 discusses the limitations of the study and possible future work and Section 9 concludes the paper.

## 2 RELATED WORK

This section discusses plane manipulation, perceptual structure of a task, and reviews various design factors considered by existing object manipulation interaction techniques in VR.

### 2.1 Plane manipulation task and perceptual structure

Object manipulation is the process of performing object translations and rotations [4]. The object is first selected, then manipulated, and finally released [4]. Plane manipulation is a type of object manipulation and we assume that the plane object represents an infinite plane. Fitts' law [15] states that the measure of difficulty is the logarithmic ratio of the distance between the hand and the target to the target size; and this applies for object manipulation in VR [34, 55]. As the target to which the plane has to be aligned is smaller, the difficulty of the plane alignment task increases making it difficult to achieve high precision.

Perceptual structure represents how a user perceives the modifiable attributes of the task [6, 39]. There are two types of perceptual tasks; integral tasks have attributes that are changed together and separable tasks have attributes that are changed independently. The perceptual structure of the interaction should match the perceptual structure of the task to achieve higher accuracy in less time. Object manipulation using controllers is an integral task but it may not be the case for precise object manipulation using hands. For precise object selection, Graham and MacKenzie observed two types of movements in object selection in VR: fast and imprecise movements as well as slow and precise movements [21]. On the other hand, Mendes et al. [35] found that their separable technique that provided small movements helped the participants to achieve higher precision than baseline in precise object manipulation but at the cost of task completion time. In this work, we explore whether hand-based plane manipulation techniques need to be integral or separable to achieve high precision in short time.

## 2.2 Design Factors for object manipulation in VR

The design factors for existing object manipulation techniques in VR have been identified to create a taxonomy as shown in Figure 1. We considered the following criteria to select interaction techniques: (1) based on either hand tracking, hand-held (controller, stylus, tracker) or gloves, (2) either user studies to evaluate these techniques, elicitation or psychology studies for object manipulation in VR, (3) published after 2000, (4) one user setup.

Generally, interaction techniques for object manipulation are either direct or indirect manipulation. In direct manipulation, objects are interacted with physically [48]. Klatzky et al. proposed a taxonomy of different gestures used for interacting with real-world objects [27] which could be used for object manipulation in VR. Pinch and grasp gestures are the most common hand gestures used for direct manipulation in VR [7, 18, 22, 37]. Pinch (precision grasp) has been found to have fewer hand tracking issues than close fist grasp (power grasp) [37].

Indirect manipulation map and transform hand movements into translation and rotation using either metaphors, widgets, or gesture mapping. Metaphor-based techniques use an analogy to help users create a mental model of the technique [14] such as handlebar [50], rails [36], spindle wheel [11], crank handle [3], knob [8], pin [20], and paper [54]. Some metaphors were hard to remember as these did not provide enough visual cues to remember and perform the necessary actions [36]. Widgets such as Smart Pin [9], separated DoF (SDOF) [35] and 7 Handle manipulation [38] have been designed. Force push [57] used gesture mapping to map different gestures to translation and rotation.

Interaction techniques that have Rotation Translation (RT) integrated such as direct manipulation have 6 DoF. When several DoF are simultaneously manipulated, a small noise in the movement might take the object far from its expected position [38]. Interaction techniques that provide RT separation [3, 8, 9, 11, 20, 35, 36, 50, 54, 57] separate translation and rotation into independent operations. Mendes et al. found that RT separation reduced unwanted transformations created in direct manipulation [36]. Interaction techniques that utilize the object axes for manipulation [3, 8, 35] have 3 DoF and this forces the user to perform the operations along with one of those specific axes. On the other hand, interaction techniques that allow users to create a custom axis for translation and rotation [9, 11, 20, 36, 42, 50, 54, 57] provide 1 DoF transformation along that custom axis. Mendes et al. [35] found that 1 DoF is helpful for fine adjustments as it constrains transformations to a single dimension and this prevents additional unwanted actions.

Most interaction techniques [3, 8, 9, 16, 35, 36, 42, 42, 54, 57] use the object center as the rotation pivot point. Some interaction techniques [11, 42, 50] create a custom pivot axis based on where the widget is activated whereas [20, 38] allowing the user to select one of the multiple handles of the widget as the pivot point.

Control-Display (CD) gain maps input device movement to display pointer movement [19]. CD gain was used for scaling translation and rotation [3, 16, 17, 35, 36, 42, 50, 57] and position and viewpoint adjustments [42]. CD gain makes object movement less sensitive to movements of hands which helps to achieve higher precision than direct manipulation [16, 17, 35].

These interaction techniques differ in terms of the hands supported. Most interaction techniques [8, 9, 16, 16, 17, 28, 35, 38, 57] used one hand, some of them [3, 11, 50] used both hands and a few [11, 20, 36, 42, 54] provided both options.

The effectiveness of feedback such as visual, haptic, auditory has been investigated for object manipulation in VR. Most interaction techniques provide visual feedback of activation and deactivation as well as other information such as transformation axes. Displaying a grid created visual disturbances in the object manipulation task [56] whereas visual feedback such as interpenetrable hands [51], semi-transparent hands [51], object silhouette [33], and semi-transparent

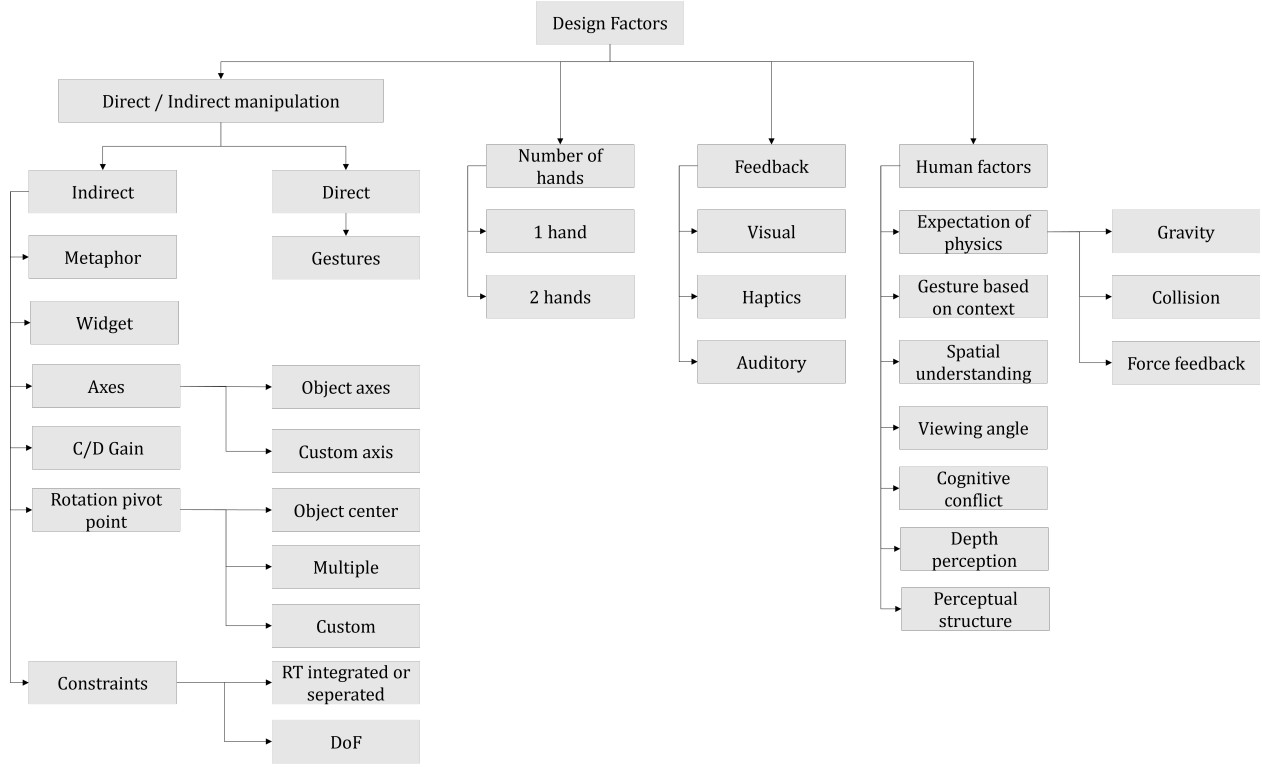

Figure 1: Proposed taxonomy of design factors for object manipulation interaction techniques in virtual reality

objects [33] improved precision. Passive haptic feedback increased the manipulation speed but reduced accuracy [56]. Audio-tactile feedback as a cue of object proximity improved spatial awareness and guided hand motion [32].

The human factors observed in user evaluation, elicitation, and psychology studies include perceptual structure [23], expectation of physics [1], preference of gestures [1], viewing angle [31], cognitive conflict between the visual and proprioception systems [49] and depth perception [43, 44]. Elicitation study [1] found that participants expected physics in the form of gravity, deformability, and contact modeling when directly interacting with the objects, and different gestures were preferred based on the context. Mapes and Moshell [31] proposed that the VR viewpoint should be moved to the most advantageous angle for direct manipulation, such user behavior was observed in [29]. According to [45], the highest precision control can be achieved on a transformation axis perpendicular to the current viewing angle. Force push [57] utilized viewing angle by restricting the user to make hand gestures along one of 3 coordinate axes based on the viewing angle.

## 3 DESIGN PROCESS OF PLANE ALIGNMENT TECHNIQUES

An iterative design process was carried out to design interaction techniques for plane alignment in VR.

First, a contextual inquiry was carried out to understand the osteotomy plane alignment step and create a task of plane alignment for the study. Next, user expectations study was conducted with controllers to understand what medical users expected while manipulating planes in VR. Potential design factors were selected from the proposed taxonomy (refer to Section 2.2) for designing separable plane alignment techniques. These interaction techniques were implemented in Unity 3D software for Oculus Quest device. Pilot tests of these interaction techniques with two HCI researchers were conducted to iterate these designs.

### 3.1 Contextual Inquiry and plane alignment task

A contextual inquiry was conducted with two experienced radiologists online. The participants were asked to explain the jaw osteotomy operation planning process using software of choice on patient data. Follow-up questions were asked to clarify details about the osteotomy plane alignment step. We found that medical users tried to position the osteotomy plane between anatomical landmarks such as two teeth and angled the osteotomy plane based on the alignment of the teeth. Based on this finding, we created the task for plane alignment in which the user has to manipulate the plane (translate and rotate) and align it between two different colored segments of a cube as shown in Figure 2(a). The task would be considered complete when a minimum acceptable accuracy level is reached as medical professionals needed to align it as precisely as possible.

### 3.2 User expectations study

A virtual reality environment was created in which the user could use direct manipulation based grasp technique with controllers to perform the plane alignment task mentioned in Sec. 3.1. The two medical professionals from the contextual inquiry participated in this study. They were asked to manipulate the plane until they felt that the plane was properly placed. They were asked open feedback about the acceptability of the using grasp with controllers for plane manipulation and their expectations. Both medical professionals felt that grasp technique of controllers was efficient to perform the task however they wanted to manipulate the plane with their hands. They wanted to use natural interaction techniques like push and poke.

### 3.3 Selection of Design Factors

Several studies [3,16,20,42,57] used direct manipulation, an integral technique, as the baseline. From the literature review, we found that separable techniques that provided users the ability to select RT and DoF were able to achieve higher precision [31, 35, 36]. Thus, we

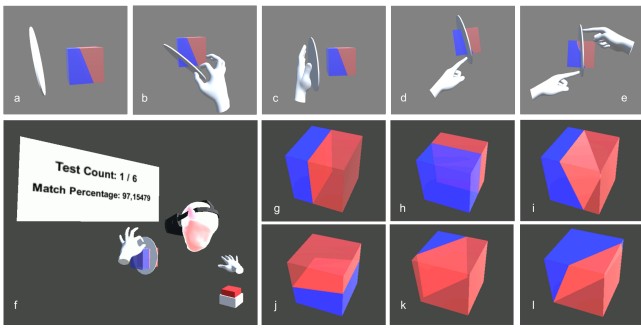

Figure 2: (a) plane manipulation task, (b) pinch, (c, d, e) Push-Poke, (f) experimental setup, (g-l) six trials of the task

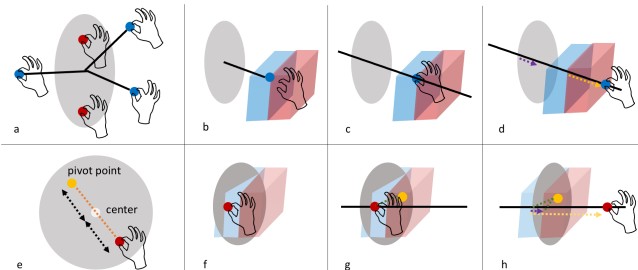

Figure 3: Custom Axis with CD Gain: (a) handles, (b) translation handle (c) translation handle pinched (d) translation handle pulled, (e) custom rotation pivot, (f) rotation handle, (g) rotation handle pinched, (h) rotation handle pulled. The dashed lines, plane center, and pivot point are for explanation and not displayed in VR.

decided to design a separable plane manipulation technique that would provide optimal RT separation and DoF selection for the user. We decided to use the design factor of custom axis as it would allow user to select RT and decide the axis for these operations. We decided to use the hand position for the direction of the custom axis for translation and rotation operations similar to Smart Pin [9]. However, in Smart Pin [9], the translation and rotation handles are centered at the object center and aligned to the object axes; and we decided to allow the user to select the two ends of the axes.

Veit et al. [52] suggested that these parameters should be dynamically selected and they selected them based on velocity but we wanted to these to be selected based on interaction. Studies have shown that humans automatically choose contact points based on their estimation of the object's center of mass while viewing the object [30]. As they interact with the object, contact points are later adjusted as humans re-evaluate the object's properties. Thus, we decided to design another interaction technique based on collision.

### 3.4 Design of interaction techniques

Interaction techniques of (1) Push-Poke and (2) Custom axis with CD gain (CACD) were designed based on the above selected design factors and later refined based on feedback from pilot tests.

#### 3.4.1 Push-Poke

From the taxonomy of hand gestures for object interaction [27], gestures of push and poke were found suitable for colliding with plane. The user can push with their palm as shown in Figure 2(c) and poke with their fingers as shown in Figure 2(d,e) to collide with the plane in zero gravity and directly manipulate it. When the user uses Push-Poke to manipulate the plane, contact normal force is applied at the contact points on the plane and the pivot point and transformation axis are selected dynamically based on the contact forces applied. For example, when the user pokes the plane with 1 finger as shown in Figure 2(d), the object rotates in 1 DoF around the pivot point which is on the opposite side of the plane similar to the rotation handle of CACD widget (explained in CACD design section below). On the other hand, if the user wants to rotate the object in 1 DoF with the pivot around the center of the object, the user can poke the object with two fingers on opposite sides as shown in Figure 2(e). This technique was implemented in Unity, the plane was modeled as a rigid body with a mass of 47.95 kg, drag of 19.6 N, angular drag of 29.76 Nm and zero gravity. Hands and cube do not have any mass. These values were initially set using trial and error and then validated through the pilot studies.

#### 3.4.2 Custom axis with CD gain (CACD)

In the first iteration, a widget based on custom axis in the direction of the hand position was designed, as shown in Figure 3. The translation and rotation handle appear based on the distance of the

dominant hand from the plane; when the user's dominant hand is near the plane, a translation blue handle appears at the dominant hand location as shown in Figure 3(b) and when the user's dominant hand is on the plane, a rotation red handle appears at the dominant hand location as shown in Figure 3(f). Once the user pinches a handle, a custom axis from the initial pinch location to the current hand location is created as shown in Figure 3(c,g), and then the user can translate or rotate along this new custom axis. When the handle is released, the operation stops then the custom axis unfreezes and the handles(s) start following the user's hand position.

The pilots revealed the usability issues with the widget. One pilot participant found it difficult to understand how to rotate the plane around the plane center and suggested that the rotation interaction should help in moving one point of the plane to the target location while keeping the opposite side of the plane fixed; "move this point there". Instead of using the object center as the pivot point for rotation, a custom pivot point is created on the opposite side of where the rotation handle is pinched on the plane as shown in Figure 3(e). This made the rotation operation work similar to the one finger poke shown in Figure 2(d). Participants found it hard to perform small movements with this technique, so a CD gain factor of 0.1 is applied to scale down the movement in translation as shown in Figure 3(d) and rotation as shown in Figure 3(h). This value was validated with pilots.

## 4 RESEARCH QUESTIONS

This section discusses the research question aimed to be answered by the user study.

**RQ1: Which interaction technique is objectively accurate, has least task completion time and most preferred for plane alignment in virtual reality?**

A certain interaction technique might be objectively more precise than others, but it might not provide a good user experience to the user. Therefore, we also focus on the user's subjective ratings to determine which interaction technique had better user experience, higher preference, and was subjectively more precise. We also focus on task completion time (efficiency) as some previous separable interaction techniques increased the time taken [3, 35, 36].

**RQ2: Which interaction technique matches the perceptual structure of a precise plane manipulation task?**

We plan to understand whether simultaneous modification of object manipulation parameters (direct manipulation), or modifying them separately through user selection or dynamic selection matches the perceptual structure of precise plane manipulation task.

## 5 EXPERIMENTAL STUDY

This section describes the experimental study in detail.

## 5.1 Participants

We recruited 12 participants (7 male, 5 female, average age 30 years, SD=5) using snowball sampling. This sample size was decided based on a power analysis calculated for repeated measures ANOVA, assuming a high effect size ($\eta^2 \geq 0.14$ or $f \geq 0.4$), a power level of 0.8, an alpha level of 0.05 and measured for 1 group and 3 measures. The effect size obtained for each quantity between the conditions is reported in the results section. Nine participants were university students and three were full-time employees including one medical professional. 11 participants had prior experience in VR and 8 participants had used hand interaction in VR. Handedness information of the participants was collected at the start of the study to determine the dominant hand for the CACD interaction technique. The participants did not receive any financial compensation for participating in the study.

## 5.2 Apparatus

The participants were asked to wear a head-mounted VR device Oculus Quest through which they performed the tasks. The headset was connected to a Schenker DTR 15 laptop using a USB-C cable. The participant was asked to do the study standing up so that they could move around the cube and observe it from different angles if required. The participants had $1x1m^2$ space around them.

## 5.3 Task

The plane alignment task is described in the design process section. To evaluate the interaction techniques for plane alignment, the cube was made stationary and the participant could only interact with the plane. The Unity3D environment contained a precut cube, a plane, a dashboard, and a button for progressing to the next trial as shown in Figure 2(f). The dashboard displayed the accuracy and the current trial number. The button was initially grey before each trial. To complete the task, the participant had to achieve at least 95% *task accuracy*. This threshold value was determined by the pilot studies; pilot participants found it hard to achieve over 95% accuracy with pinch. This threshold value required a minimum number of interactions with all the interaction techniques which would give the participants enough time to get a better understanding of using the interaction techniques.

The *task accuracy* was calculated using Eqn. 1 as a weighted sum of *distance accuracy* (*DA*) and *angle accuracy* (*AA*). The *plane distance* was computed as the Euclidean distance from the plane center to the closest point on the ground truth plane (plane used to cut the cube into two segments) and was used to compute the *DA* as mentioned in Eqn. 2. The *angle distance* was computed as the Euclidean distance between the end points of plane's unit normal vector and the ground truth plane's unit normal vector starting from the same point. The ground truth plane's unit normal was flipped in case the normal was pointed in the wrong direction in Unity. This *angle distance* was used to calculate the *AA* as mentioned in Eqn. 3.

Although the user could manipulate the plane in 6 DoF; the *accuracy* function was calculated in 3 DoF; the plane was determined by a direction (the normal vector, which has 2 DoF, ignoring the rotation around the normal), plus an offset or perpendicular distance from the origin (adding a 3rd DoF, ignoring the translation perpendicular to the normal). The button became red when the threshold for accuracy was reached. Participants could press the red button to advance to the next trial in the condition.

$$accuracy = \frac{DA + 2 \times AA}{3} \quad (1)$$

$$DA = \begin{cases} 0 & \text{plane distance} > 0.35 \\ 100 - \frac{\text{plane distance} \times 100}{0.35} & \text{otherwise} \end{cases} \quad (2)$$

$$AA = \begin{cases} 0 & \text{angle distance} > 1 \\ 100 - \text{angle distance} \times 100 & \text{otherwise} \end{cases} \quad (3)$$

## 5.4 Design

We needed direct manipulation as a baseline as we needed to compare our techniques to a technique that simultaneously changes the object manipulation parameters (as explained in RQ3). We decided to use pinch as baseline because it has the least hand tracking errors out of the commonly used hand gestures for direct manipulation [37]. We used within-subject evaluation to compare the different interaction techniques.

- Pinch (baseline)

- Push-Poke

- Custom axis with CD gain

In the study, the participants were presented with conditions in counterbalanced order based on Balanced Latin Square to reduce the effect of ordering the conditions. A total of six trials are used in each condition and the precut cubes used for these trials are shown in Figure 2(g-l). The plane was not reset between the trials. The initial plane was placed and the six cubes were cut such that the minimum accuracy at the start of each trial was less than 35% and the user had to transform the plane along at least one coordinate axis. Overall, the study took around 45 minutes to complete.

While the participants were performing the task, measures such as task completion time (TCT), accuracy, distance accuracy, angle accuracy, interaction count, interaction type (translation or rotation for CACD), start time, and end time of each interaction were collected. Execution time is calculated as the duration of an interaction and evaluation time is calculated as the time between when an interaction finishes and when the next interaction starts. These are used to represent the decision and response time for each interaction event for the objective calculation of cognitive load [2, 12]. To understand the types of movements offered by these interaction techniques, we calculate the minimum and maximum movements made by these interaction techniques. Since the movement can be computed in terms of position and orientation, accuracy changes (which uses differences in both position and orientation between ground truth plane and current plane) are used as an indicator of movement. The consecutive accuracy changes of more than 0.01 are considered in this calculation to ignore the noise. We calculated the median of these measures across trials instead of mean to remove the effect of outliers.

We asked the participants about their subjective ease of use, learnability, confidence, hand tiredness, intuitiveness, precision, use daily of the conditions on a Likert scale from 1 to 7. These subjective ratings are taken from the previous evaluations of object manipulation techniques in VR [3, 8, 9, 11, 16, 20, 20, 28, 36, 36, 38, 50, 57] and System Usability Scale (SUS) [5]. In addition, for Push-Poke, the participants were asked to rate the ease of use of push and poke separately and translation and rotation handles of CACD. The participants were asked to clarify the subjective rating of 4 and below. They were also asked open-ended questions about the positives and negatives of the conditions to gain further understanding of their perceptions. After completing the three conditions, the participants were asked to rank the interaction techniques from 1 to 3 (1 is the highest rank and 3 is the lowest rank) on the aspects of preference, most precise, best suited for the novice user, and the most potential to be developed further similar to previous studies asking to rate preference [9, 11, 16, 24, 28, 38, 42].

### 5.5 Procedure

The procedure of the controlled experiment was as follows:

*1) Welcome and demographics.* Due to the COVID-19 situation, appropriate precautionary measures were taken such as cleaning the VR device, computer, desk, and chair with disinfectant before and after each study. The recruited participants were informed before the study about the process and the safety measures that were taken. The participants and the moderator wore masks and kept their distance throughout the session. The participants were initially welcomed and were briefed on the purpose of the study, the task, interaction techniques, and data collection. The participants were made aware that the aim of the study was accuracy and not speed so that they focus on accuracy. The participants were informed that they could discontinue the study whenever they wanted for any reason including VR sickness. The participants were asked to sign the consent form. After this, background information such as age, gender, profession, dominant hand, and their experience in using VR was collected.

*2) Training.* Before the actual tasks, the participants were provided with training to get familiar with the interaction technique and task. The training task was similar to the real task, however, the precut plane was randomly generated. The participant could take as much time and trials to practice. The participant could press the button to practice with another randomly generated plane. The moderator observed the VR view on the laptop screen in case the participants needed help.

*3) Interaction.* We asked participants to align the plane as described above. After completing each condition, the participants were asked to rate the condition on the subjective measures. Based on the ratings, a semi-structured interview about the reasons for rating as well as positives and negatives opinions of the interaction techniques were asked.

*4) Survey.* After completing the three conditions, the participants were asked to rank the interaction techniques.

### 5.6 Analysis

One-way repeated measures Analysis of Variance (ANOVA) was used to find a significant difference in measures between the three conditions when the distribution was normal, had equal variances and the sphericity condition held. Posthoc paired sample t-test was performed using Bonferroni Correction with a corrected p-value of $.05/3 = .0167$ to find the significant difference between pairs of conditions and we calculated the Cohen's d as a measure of the effect size. If the above conditions did not hold, we used the non-parametric Friedman Chi-Square Test to find the significant differences in the objective measures between the conditions and posthoc Wilcoxon signed-rank Test with Bonferroni correction was performed to find the significant differences between pairs of conditions and we calculated the matched pairs ranked-biserial correlation ($r$) values as a measure of the effect size.

## 6 RESULTS

This section presents the results of the study.

### 6.1 Objective measures

The distribution of the objective measures collected during the study is shown in Figure 4. These objective measures showed a significant difference between the conditions (p values for between condition comparison are shown in Figure 4 and p values for pairwise comparison are shown in Tab. 1 and Tab. 2). Posthoc tests showed that Push-Poke was significantly more accurate than pinch in terms of overall, distance and angle accuracies. It took significantly more time to complete the task with CACD than the other two techniques, and participants interacted significantly less times with pinch than the other two techniques. The minimum and maximum accuracy changes with pinch were higher than Push-Poke which in turn were higher than CACD. Participants took significantly less evaluation time with

Push-Poke than the other two techniques whereas participants took significantly more execution time with Pinch in comparison to other two techniques.

Accuracy changes across the trial duration are plotted to understand the types of movement (small or large) made during the task, which we refer to as the accuracy trend. Accuracy changes over 70% are considered for calculating the trend lines, as the starting accuracy for different trials might be different. The task time is normalized for all trials then accuracy changes are modeled by Support Vector Regression (SVR) with a radial basis function (RBF) kernel with $C = 10$, $\gamma = 0.1$, and $\varepsilon = 0.1$. Figure 4(k,l,m) shows the accuracy trend across the trials for three interaction techniques for each participant. Figure 4(j) shows the accuracy trend across participants for each of the interaction techniques.

### 6.2 Subjective Data

The distribution of the subjective ratings for each condition is shown in Figure 5. Friedman Chi Square Test showed a significant effect of condition all subjective ratings except ease of use for components (p values for between condition comparison are shown in Figure 4 and p values for pairwise comparison are shown in Tab. 1 and Tab. 2). Posthoc tests showed that Push-Poke was significantly more easy to use, intuitive, confident using it and prefer to use it daily than both pinch and CACD. Push-Poke was significantly easier to learn and not tired than CACD and significantly more precise than pinch. Pinch was significantly intuitive than CACD.

*1) Learnability:* The ranking of conditions based on their suitability for novice users is mentioned in Tab. 4. Participants felt the familiarity of pinch and Push-Poke gestures in real life would help novices in learning. Some participants faced difficulties in learning CACD especially in using handles, translation on a custom axis, and understanding CD gain. P8 said "use of the handles was somewhat different and not so familiar" and P11 said "rubber band effect takes some time to get used to". P12 said "learning translation was hard. Movement along one axis (x, y, z) at a time is easy. Moving along a combination of multiple axes is hard".

*2) Ease of use:* Participants faced problems in determining when to release the plane in pinch interaction, P11 said "there was a slight delay when releasing the object which caused it to misalign several times" and P2 said, "it is hard to understand when to let go". Sometimes participants were not able to pinch the handles in CACD. For Push-Poke, participants reported it was hard to Push-Poke behind the plane as the hands were being occluded by the plane. Participants overcame this by moving around to view it from a different angle. They also had to lower the speed for push as hand movements above a certain speed were not detected. Friedman Chi-Square tests showed no significant difference in ease of use between Push-Poke and its components as well as CACD and its components.

*3) Hand tiredness:* Some participants felt pinch tired their hands as they had to hold the plane for a long time before releasing it. P8 said "It was difficult to place the plane precisely. My hands got tired as I was grabbing the object continuously". Participants felt their hands got tired with CACD because of the number and amount of movements needed for CD gain.

*4) Confidence:* Participants felt they should be able to move the plane in one interaction with pinch, P12 said "I know how to move it, but it is hard to get it right on the first trial". The improper release functionality caused the pinch technique to be more challenging. Some participants took time to learn CACD and later became confident in using it, P8 mentioned "I was very confident in the last exercises and my aim was more precise".

*5) Precision:* The subjective ranks of the conditions in terms of precision are listed in Tab. 4. Participants mentioned Push-Poke needed less effort to precisely manipulate objects. Few participants felt the rotation handle of CACD allowed them to make precise movements; P12 remarked, "rotation in custom axis gives more

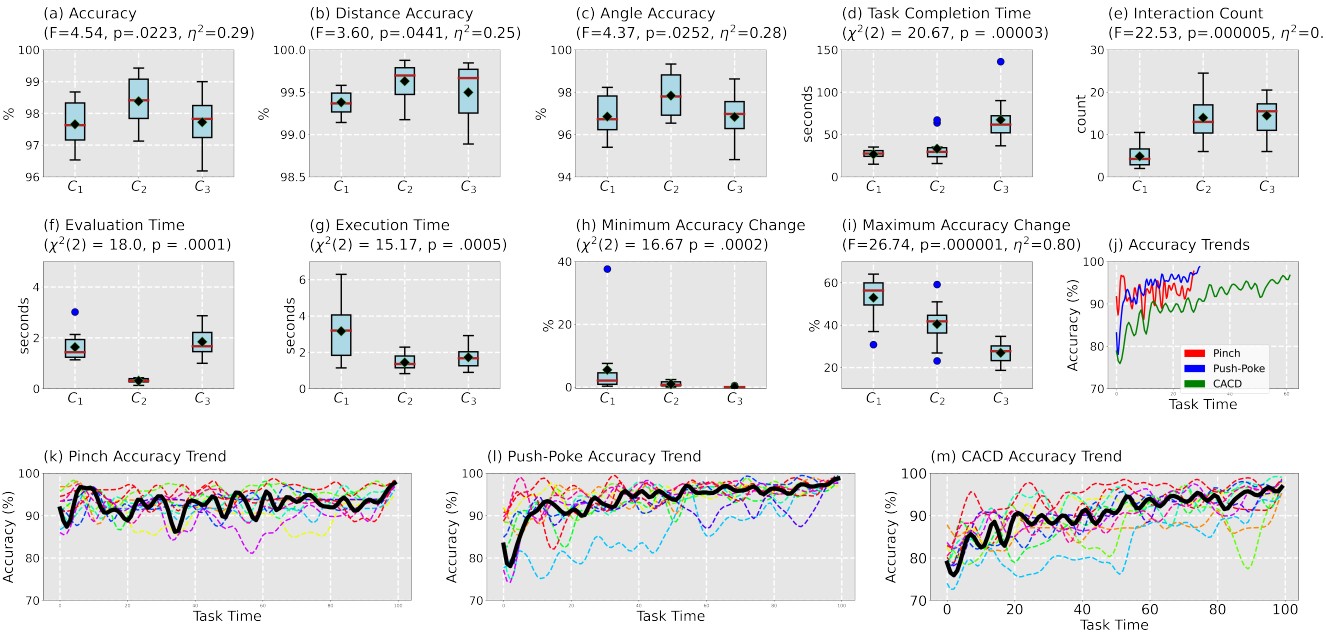

Figure 4: Quantitative evaluation results: (a) distribution of the median accuracy for each condition, (b) distribution of the median distance accuracy for each condition, (c) distribution of the median angle accuracy for each condition, (d) distribution of median TCT for each condition, (e) distribution of median interaction count for each condition (f) distribution of the median of median evaluation time for each condition, (g) distribution of the median of median execution time for each condition, (h) distribution of the median of minimum accuracy change, (i) distribution of the median of maximum accuracy change, (j) accuracy trend for each condition scaled by median task completion time, (k) accuracy trend of a pinch for each participant, (l) accuracy trend of Push-Poke for each participant, (m) accuracy trend of CACD for each participant. Box plot comparison of the 3 conditions ($C_1$) pinch, ($C_2$) Push-Poke, ($C_3$) CACD are shown.

control for small movements".

*6) Preference:* Participants' ranking of conditions based on their preference is listed in Tab. 4. Many participants explained that ease of use and precision were used to determine the preference. While ranking their preference, participants mentioned Push-Poke felt accurate, natural to use and was possible to use two hands. Pinch was easy to use but it did not feel accurate. Custom axis + CD Gain felt accurate, but it was difficult to use.

*7) Most potential for future development:* Participants' ranking of conditions based on their potential for future development is listed in Tab. 4. Participants found the Push-Poke condition fun and wanted to see how it can be developed further. Participants suggested pinch to be made a two-handed operation to offer more control, P4 suggested "to lock other side while rotating like the rotation in CACD method" and P10 suggested to "stabilize one edge with my left hand and at the same time make a rotational movement with my right hand". Participants suggested combinations of techniques: (1) pinch and poke, (2) pinch with rotation handle with CD gain.

## 7 DISCUSSION

The results indicate that Push-Poke was objectively and subjectively more precise than the baseline pinch. Participants felt it was easy to use, intuitive, felt confident in using it than pinch and CACD. Participants also felt it was easier to learn and less tiring than CACD. Based on these results, the perceptual structure of hand based plane manipulation in VR is discussed below.

### 7.1 Perceptual structure of precise plane manipulation task consists of integral and separable tasks

As seen in Tab. 2, the minimum accuracy change of CACD and Push-Poke are significantly smaller than pinch. The overall trend lines in

Figure 4(j) show that participants were able to make small accuracy improvements across time using Push-Poke and CACD. This is also supported by participants' quotes; P12 said "rotation in custom axis gives more control for small movements" while P8 said, "Push-Poke gave me the freedom to correct the position". This corroborates with [35] which suggests interaction techniques should support small movements for precise manipulation. This is also supported by participants' suggestion to make pinch offer more control over small movements by making it a two-handed operation.

Although plane manipulation techniques must provide small precise movements, participants felt these should also support large movements. Participants suggested combinations of techniques so that these support both small and large movements such as (1) one and two handed pinch, (2) pinch and poke and (3) pinch and rotation handle with CD gain. These suggestions corroborate with Graham and MacKenzie [21] in which the movement can be divided into two phases: (1) initial fast and imprecise movement to the target and (2) final slow and precise movements. In the interaction techniques suggested by participants, the pinch interaction is more suitable for large movements whereas the poke and rotation handle are suitable for small movements. Users might have preferred direct manipulation-based pinch for large movements as it would require fewer interactions to manipulate the plane and place it near the ground truth plane. Therefore, the first large movements in plane manipulation are an integral task. On the other hand, participants preferred interaction techniques that are separable for precise movements as they provide control over plane manipulation parameters. These interaction techniques helped in achieving the last few % increments in accuracy. Thus, precise plane manipulation could be split into two perceptual tasks: initial large movements as integral tasks and final precise movements as separable tasks.

Table 1: Results of pairwise t-Tests with Bonferroni Correction between the pairs of conditions. Statistically significant results are reported as p < .001/3***, p < .01/3**, p < .05/3*. † indicates the opposite comparison statistics have been reported.

| quantity | Comparison 1 Push-Poke > pinch | | Comparison 2 Push-Poke > CACD | | Comparison 3 CACD > pinch | |
|---|---|---|---|---|---|---|
| | $p$ | $d$ | $p$ | $d$ | $p$ | $d$ |
| Accuracy | .0001*** | 0.97 | .0359 | 0.81 | .4141 | 0.08 |
| Distance Accuracy | .0026** | 1.38 | .0654 | 0.43 | .1735 | 0.41 |
| Angle Accuracy | .0006** | 0.96 | .0301 | 0.89 | .4849† | 0.01 |
| Interaction Count | .0001*** | 2.08 | .61 | 0.10 | < .0001*** | 2.62 |
| Maximum Accuracy Change | .0079*† | 1.22† | .0005** | 1.73 | < .0001***† | 3.17† |

Table 2: Results of Wilcoxon signed-rank tests with Bonferroni Correction between the pairs of conditions. Statistically significant results are reported as $p < .001/3***$, $p < .01/3**$, $p < .05/3*$. † indicates the opposite comparison statistics have been reported.

| Quantity | Comparison 1 pinch > Push-Poke | | | Comparison 2 CACD > Push-Poke | | | Comparison 3 CACD > pinch | | |
|---|---|---|---|---|---|---|---|---|---|
| | $W$ | $p$ | $r$ | $W$ | $p$ | $r$ | $W$ | $p$ | $r$ |
| Task Completion Time | 62.0† | .0386† | 0.59† | 78.0 | .0002*** | 1.0 | 78.0 | .0002*** | 1.0 |
| Evaluation Time | 78.0 | .0002*** | 1.0 | 78.0 | .0002*** | 1.0 | 49.0 | .2349 | 0.25 |
| Execution Time | 77.0 | .0005** | 0.97 | 57.0 | .0881 | 0.46 | 78.0† | .0002***† | 1.0† |
| Minimum Accuracy Change | 70.0 | .0061* | 0.79 | 74.0† | .0017**† | 0.9† | 78.0† | .0002***† | 1.0† |

## 7.2 Interaction techniques for small precise movements should dynamically select plane manipulation parameters based on contact point(s)

The rotation handle of CACD dynamically selects RT, DoF, transformation axis and pivot point based on where the handle is grabbed whereas Push-Poke dynamically selects RT and DoF as well as pivot point, and transformation axis based on contact point(s). The Push-Poke technique helped participants to reduce perceivable plane misalignment with confidence. P12 said "I can see the error; I can work on that particular error to go away". Participants felt rotation was easy to perform in CACD which contradicts with other indirect manipulation techniques [3, 17, 35, 36, 50]. Mendes et al. [36] explained that participants required better understanding and experience of rotation axes to perform rotation around a custom axis. The results showed our design of dynamically selecting plane manipulation parameters based on contact points in CACD is inline with the user's mental model.

## 8 LIMITATIONS AND FUTURE WORK

During the study, participants were asked to take their time to learn and practice the CACD technique. Some participants felt they needed more time to learn CACD effectively. The results of the study may differ if more time is provided to participants to practice this interaction technique. In addition, longitudinal studies could be performed to observe any changes in task performance and user experience over time.

These interaction techniques could be improved based on the participants' feedback. Participants wanted to move the plane at a faster speed using push interaction. The physics settings could be changed so that Push-Poke could support a larger speed of movement which may impact the accuracy trend and the final precision achieved. CACD could support both hands so that the user can use the closest hand for interacting with the handles. Variable CD gain could be used in CACD to reduce hand tiredness. Based on the participants' suggestions, combinations could be designed such as (1) one and two-handed pinch, (2) pinch and poke, (3) pinch and rotation handle of CACD. In addition, haptic, auditory or visual feedback could be added for indicating states such as when the plane is touched, interaction has started, and stopped. Also, hands could be made transparent to address the problem of occlusion.

These proposed interaction techniques could be further extended and validated for jaw osteotomy planning with medical professionals. Medical professionals might want to manipulate the skull and plane individually and together. These designed interaction techniques could be adopted and extended to manipulate 3D models such as skulls. Push-Poke could be used to push and poke the bounding box around 3D models. The handles of CACD can be created on the closest surface of the bounding box around 3D models. Future work could evaluate these proposed interaction techniques to manipulate 3D models for jaw osteotomy planning with medical professionals and evaluate them against controllers.

## 9 CONCLUSION

In this paper, we investigated the perceptual structure of hand-based plane manipulation. Several interaction techniques were designed to provide precision in presence of noisy hand tracking by considering object manipulation as a separable task; by providing users with control over the object manipulation parameters. We proposed two interaction techniques for hand-based plane manipulation in VR: Push-Poke which dynamically selected these parameters and CACD that allowed users to select these parameters. A controlled study was conducted in which these separable interaction techniques were evaluated with a pinch as a baseline for a representation of integral interaction technique.

The results revealed that the Push-Poke interaction technique was objectively and subjectively more precise and preferred because it is intuitive, easy to use and participants felt more confident while using it. From results of this study, we found that hand-based precise plane manipulation is composed of integral and separable tasks: integral task for initial large movements and separable task for final small precise movements; and the technique for small precise movements should dynamically select plane manipulation parameters based on contact point(s). Future work could implement the suggested combination of interaction techniques for 3D object manipulation. Studies with medical professionals for jaw osteotomy operation planning could be conducted to validate whether the proposed interaction techniques can overcome noisy hand tracking so that hand tracking could potentially replace controllers.

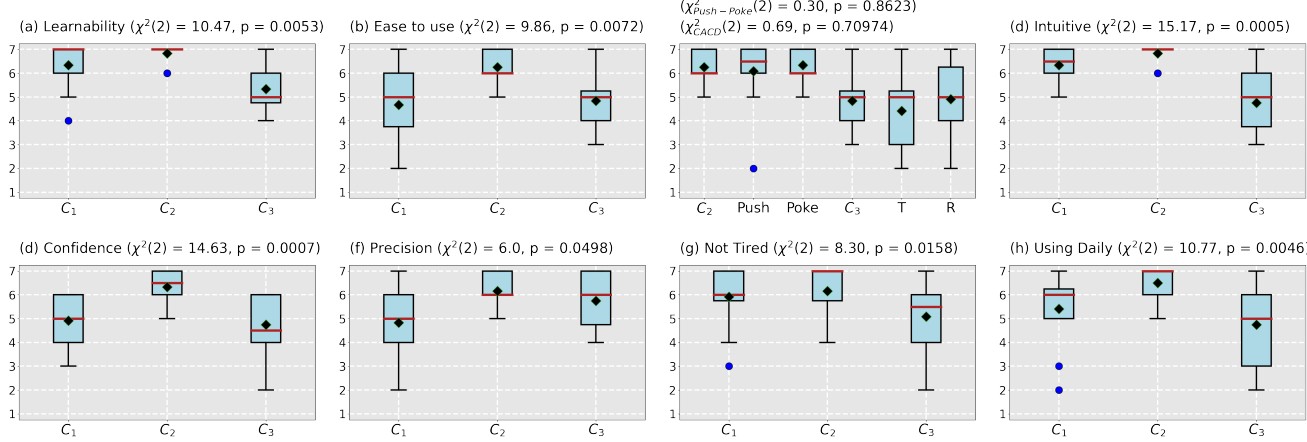

Figure 5: Box plots of subjective ratings of each item according to each condition: ($C_1$) pinch, ($C_2$) Push-Poke, ($C_3$) CACD, (T) translation handle in CACD, (R) rotation handle in CACD, and result of Friedman Chi-Square Test. The blue circles represent the outliers, black diamonds represent the mean and the red lines represent the median.

Table 3: Results of Wilcoxon signed-rank tests with Bonferroni Correction between the pairs of conditions. Statistically significant results are reported as $p < .001/3$***, $p < .01/3$**, $p < .05/3$*. † indicates the opposite comparison statistics have been reported.

| quantity | Comparison 1 Push-Poke > pinch | | | Comparison 2 Push-Poke > CACD | | | Comparison 3 pinch > CACD | | |
|---|---|---|---|---|---|---|---|---|---|
| | $W$ | $p$ | $r$ | $W$ | $p$ | $r$ | $W$ | $p$ | $r$ |
| Learnability | 60.0 | .0480 | 1.0 | 73.0 | .0040* | 1.0 | 62.5 | .0342 | 0.71 |
| Ease of use | 72.0 | .0052* | 0.91 | 73.0 | .0038* | 0.85 | 42.5† | .4060† | 0.91† |
| Intuitive | 67.5 | .0120* | 1.0 | 76.5 | .0018** | 1.0 | 71.0 | .0063* | 0.83 |
| Confidence | 77.5 | .0012** | 1.0 | 75.0 | .0025** | 0.10 | 44.5 | .3450 | 0.18 |
| Precision | 71.0 | .0061* | 0.87 | 49.0 | .2260 | 0.38 | 59.0† | .0615† | 0.58† |
| Not tired | 44.5 | .3442 | 0.27 | 70.5 | .0070* | 1.0 | 60.5 | .0479 | 0.64 |
| Using daily | 70.5 | .0069* | 1.0 | 73.0 | .0041* | 1.0 | 50.5 | .1922 | 0.27 |

Table 4: Accumulated count of rankings

| Accumulated count | Condition | Rank 1 | Rank 2 | Rank 3 |
|---|---|---|---|---|
| **Suitable for novice user** | Pinch | 4 | 7 | 1 |
| | Push-Poke | 8 | 4 | 0 |
| | CACD | 0 | 1 | 11 |
| **Precision** | Pinch | 0 | 4 | 8 |
| | Push-Poke | 10 | 2 | 0 |
| | CACD | 2 | 6 | 4 |
| **Preferred** | Pinch | 0 | 8 | 4 |
| | Push-Poke | 11 | 1 | 0 |
| | CACD | 1 | 3 | 8 |
| **Future development** | Pinch | 2 | 8 | 2 |
| | Push-Poke | 8 | 2 | 2 |
| | CACD | 2 | 2 | 8 |

## ACKNOWLEDGMENTS

The authors wish to thank the dentomaxillofacial radiologists that provided the knowledge of their specialty. This work has been funded by Business Finland, project Digital and Physical Immersion in Radiology and Surgery (decision number 930/31/2019).

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
