# OpenReview forum: "Push-Poke: Collision based Direct Manipulation Technique for Plane Alignment in Virtual Reality"
_graphicsinterface.org/Graphics_Interface/2022/Conference — GI 2022_

### Official Review · Reviewer_Ya8o · 2022-01-14
**Interesting work to understand and test a specialized technique**

**Rating:** 7
**Confidence:** 4

**Review:**

The paper presents a design space for plane-manipulation techniques in VR, selects three techniques for comparison, and reports on a comparative study. The Push-Poke technique allows users to separate movement and tilting actions, and was shown to be more accurate than a Pinch-based technique.

Overall the paper is well grounded in the design-space work, and although there are limitations to the study, the paper provides a useful increment in knowledge about specialized interaction techniques for VR.

There are however several improvements that should be made to the paper. First, the paper desperately needs images to illustrate and explain how the different techniques work, and what the experimental tasks entailed. Second, the authors should include proxy-based techniques in their design space (e.g., manipulating the plane using a physical object as in Issartel, Paul, Florimond Guéniat, and Mehdi Ammi. "Slicing techniques for handheld augmented reality." 2014 IEEE symposium on 3D user interfaces (3DUI). IEEE, 2014.) Third, the authors should make more use of the Jacobs paper on integrality/separability, as this is a major factor in the results.

---

### Official Review · Reviewer_JYe9 · 2022-01-15
**Issues with writing; non-standard metrics. Motivation for the task not clear.**

**Rating:** 2
**Confidence:** 5

**Review:**

the paper is presents a study in which 3 different 3D object manipulation techniques are presented. While the task of 3D object manipulation is obviously important, the paper contains numerous flaws. Plain manipulation could be a 5 DoF task (when the plane is finite). However, the metrics for accuracy used in the paper do not consider the fact that the task involves essentially a disk (at least, it's not clear of the centre of the disk needs to be close to the centre of the cube); this effectively makes the task 4DoF. The metrics chosen - "accuracy" - are rather non-standard. Euclidean distance normalized to object width/screen width, or similar, could be used as a distance error (rather than the "accuracy") metric. The angle between two vectors could then be used as the other error metric.

Now, if the task is a 4DoF, does the benefit of using a 3D manipulation device still apply? Why was mouse not considered as an alternative (using the best mouse-based technique found for a similar task)? Also, if input noise is a factor, was filtering ever considered (e.g., even as simple as just averaging 10 samples)?

The stats tests are presented in a fashion different from the accepted practice (Figure 4): e.g., for p values below some chosen threshold (0.05, 0.001, etc.) one reports them as "p < .05" (the leading zero is optional here)

Some of the data in the graphs is probably unnecessary (e.g, the interaction count).

Greek letters gamma and epsilon (and others) should be typeset accordingly. >= and x instead of a proper multiplication symbol look unprofessional.

Unfortunately, the paper contains too many flaws for me to be able to recommend for its acceptance.

---

### Official Review · Reviewer_gzXS · 2022-01-17
**positioning and orienting a plane in VR**

**Rating:** 5
**Confidence:** 3

**Review:**

The submission focuses on the question of how to allow a user to position a plane, in VR, using bare hand gestures.  This task is motivated by a need in osteotomy planning to specify a cutting plane through bone, and the use of bare hand gestures is motivated by a 'contextual inquiry', as well as previous work (introduction, 2nd paragraph), that indicates that users prefer using their hands without additional hardware.
A novel interaction technique called "push-poke" is proposed.  The idea in "push-poke" is that the plane can be pushed by the palm (Figure 2c), or pushed by a single finger (Figure 2d), or by two fingers (Figure 2e).  This interaction technique was experimentally compared against two other interaction techniques ('pinch', Figure 2b, where the user uses a pinch gesture to grab the plane and then translate and rotate it; and 'Custom Axis with [low] CD Gain' = 'CACD', where the user uses a pinch gesture to either grab a point near the plane's surface to rotate the plane, or uses a pinch gesture to grab a point far from the plane which then defines an axis for translating the plane).  The experiment involved 12 users, and found that the proposed "push-poke" technique outperformed the other two techniques.

Much of the submission is well written and follows a reasonable methodology.  Some weaknesses, which I describe next, make me hesitate to recommend publication in its current form.

I can imagine two relevant contexts where a user could position and orient a plane in 3D.  The first is on a desktop (or laptop) computer, with a monoscopic display.  The positioning and orientation of a plane can be done in such a context using commonly available input devices, namely a mouse and a keyboard, by virtue of 3D widgets, as found in software like Autodesk Maya or Blender.  The user can rapidly switch between camera navigation (to look at a jawbone from different points of view) and plane manipulation using keyboard shortcut keys.  A second relevant context is in VR with a headset, where benefits could include enhanced depth perception thanks to stereo display and motion parallax (due to small and frequent head motions), as well as the benefit of the ability to use two hands simultaneously, each with 6 degrees of freedom.  It may be true that medical professionals are willing to tolerate the inconvenience of wearing a VR headset for such benefits.  However, I am not convinced that the same medical professionals will tolerate wearing a VR headset and then also insist on using bare hand gestures (which the submission admits are noisy) rather than using handheld hardware controllers that allow for more precise, reliable tracking and multiple, unambigously detected buttons press events.  Such controllers would allow for rapid, simultaneous rotation and translation of both the 3D model (skull, jawbone, etc.) and the plane, as well as the scaling up or down in size of the 3D model by pressing a different button and moving hands apart or closer together.  The video at https://www.youtube.com/watch?v=jnqFdSa5p7w  gives some idea of how rapid, natural, and expressive it can be to use two controllers to manipulate a 3D model (notice the left hand is often moving the 3D model, while the right hand applies some operation).  The experiment in the submission did not compare the proposed technique with a monoscopic desktop PC + mouse (which is a more convenient hardware platform) nor with a VR + hardware controllers configuration (which I predict would be more effective).  I am not convinced by the 'contextual inquiry' mentioned in the submission's introduction that medical professionals would prefer bare hand gestures, since these potential users were presumably not given a chance to try the task with handheld controllers.  The submission's introduction mentions that 'medical professionals ... prefer to use their hands so they do not need to learn [how to use additional hardware]', but the use of bare hand gestures also requires learning those gestures.  Another great example of a two-handed user interface, using handheld hardware devices, for positioning and orienting a plane, is found in the highly cited work of Hinckley et al. from 1996: https://scholar.google.com/scholar?q=hinckley+Passive+real-world+interface+props+for+neurosurgical+visualization

Two more reasons for me to hesitate are some unclear things about the experiment:

=> A plane in 3-space has 3 degrees of freedom.  This can be seen in the equation for a plane, ax + by + cz + d = 0, where (a,b,c) is the normal vector which has 2 degrees of freedom, and d adds a 3rd degree of freedom.  Another way to think of this is that a plane is determined by a direction (the normal vector, which has 2 DoF), plus an offset or perpendicular distance from the origin (adding a 3rd DoF).  Another way to see this is that a plane defined by a point (+3 DoF) and orientation (+3 DoF) remains invariant under rotation around its normal (-1 DoF) and also invarant under translations perpendicular to the normal (-2 DoF), and adding these up yields +3+3-1-2 = 3 DoF.  However, the submission incorrectly states (in section 2.1, 1st paragraph) that a plane has 5 DoF, and quantifies the accuracy of a plane as a function of the distance between the plane's center position and the desired center position (section 5.3, and equation 2) even though any displacement that is parallel to a plane does not change how the plane partitions space.  I do not understand why the 'distance accuracy' in equation 2 was used in the experiment.

=> The 'task completion time' (TCT), 'execution time', and 'evaluation time' mentioned in section 5.4 (3rd paragraph) and shown in figure 4 are not clearly defined.  Is TCT the total time to perform six trials (mentioned in section 5.4, 2nd paragraph)?  Should we expect TCT = execution time + evaluation time?  Figure 4e shows a median execution time for C3 as less than 2 seconds, and Figure 4d shows median evaluation time for C3 around 2 seconds, but these do not add to the median task completion time of more than 50 seconds for C3 in Figure 4b.

---

### Decision · Program_Chairs · 2022-01-18

Accept